# Wave-induced loss of ultra-relativistic electrons in the Van Allen radiation belts

Yuri Y. Shprits[1,2,3], Alexander Y. Drozdov[2], Maria Spasojevic[4], Adam C. Kellerman[2], Maria E. Usanova[5], Mark J. Engebretson[6], Oleksiy V. Agapitov[7,8], Irina S. Zhelavskaya[1], Tero J. Raita[9], Harlan E. Spence[10], Daniel N. Baker[5], Hui Zhu[2] & Nikita A. Aseev[1]

The dipole configuration of the Earth's magnetic field allows for the trapping of highly energetic particles, which form the radiation belts. Although significant advances have been made in understanding the acceleration mechanisms in the radiation belts, the loss processes remain poorly understood. Unique observations on 17 January 2013 provide detailed information throughout the belts on the energy spectrum and pitch angle (angle between the velocity of a particle and the magnetic field) distribution of electrons up to ultra-relativistic energies. Here we show that although relativistic electrons are enhanced, ultra-relativistic electrons become depleted and distributions of particles show very clear telltale signatures of electromagnetic ion cyclotron wave-induced loss. Comparisons between observations and modelling of the evolution of the electron flux and pitch angle show that electromagnetic ion cyclotron waves provide the dominant loss mechanism at ultra-relativistic energies and produce a profound dropout of the ultra-relativistic radiation belt fluxes.

[1] Helmholtz Centre Potsdam, GFZ, German Research Centre For Geosciences, Section 2.3, Building K 3, Room 012, Potsdam 14467, Germany. [2] Department of Earth, Planetary, and Space Sciences, University of California, Los Angeles, California 90095-1567, USA. [3] Universität Potsdam, Institut für Physik und Astronomie, Potsdam 14476, Germany. [4] Department of Electrical Engineering, Stanford University, Stanford, California 94305-9505, USA. [5] Laboratory for Atmospheric and Space Physics, University of Colorado Boulder, Boulder, Colorado 80303, USA. [6] Physics Department, Augsburg College, Minneapolis, Minnesota 55454, USA. [7] Space Sciences Laboratory, University of California, Berkeley, California 94720, USA. [8] Natilonal Taras Shevchenko University of Kyiv, Kyiv 01033, Ukraine. [9] Sodankylä Geophysical Observatory, Sodankylä, Finland and University of Oulu, Oulu 99600, Finland. [10] Institute for the Study of Earth Oceans and Space, University of New Hampshire, Durham, New Hampshire 03824-3525, USA. Correspondence and requests for materials should be addressed to Y.Y.S. (email: yshprits@gfz-potsdam.de).

The dynamic evolution of the Earth's electron radiation belts is the result of the competition between acceleration and loss processes. Recent observations and modelling provided significant advances in our understanding of acceleration mechanisms operating in the Earth radiation belts[1–3], whereas the loss mechanisms remain more controversial. Understanding of acceleration provides only 'half' of the global picture required for understanding the evolution of electrons, as the dynamics of the radiation belts are not determined by just the acceleration mechanisms, but rather the dynamic battle between acceleration and loss.

The earliest studies of radiation belt dynamics noted that electron fluxes tend to decrease during the main phase of geomagnetic storms[4] and these flux dropouts were attributed to reversible processes[5]. That is, when slow changes occur in the Earth's magnetic field configuration during geomagnetic storms, electron fluxes are redistributed in radial distance, energy and pitch angle (the angle between the electron velocity vector and the magnetic field). When the magnetic field relaxes back to the pre-disturbance configuration, the electron distribution also reverses to the original state. Therefore, no net loss of electrons occurs. Recent studies[6–8] showed that non-reversible electron dropouts occur during the main phase of many storms. One plausible theory to explain the non-reversible dropout is scattering by electromagnetic ion cyclotron (EMIC) waves[6,9,10], which are often strongly enhanced during geomagnetic storms. After scattering by the waves, the electrons are lost as a result of precipitation into the upper atmosphere. Another mechanism to explain the dropouts has been proposed[11]. In this mechanism, compression of the outer boundary of the Earth's magnetosphere during geomagnetic storms causes electrons at large radial distances to be lost to the interplanetary medium. Such loss creates steep radial gradients in electron-phase space density (PSD) that drive outward radial diffusion. The outward radial diffusion transports particles away from the Earth into the region of lower magnetic field and decelerates particles, which in turn propagates the electron loss to lower radial distance.

To differentiate between the two non-reversible processes, it is necessary to examine changes in the electron energy spectrum and pitch angle distribution. Numerical modelling has shown that dropouts observed down to low energy (100's of kiloelectron volts) cannot be produced by EMIC waves, but are consistent with expectations from loss produced by outward radial diffusion[11]. Additional strong evidence for the outward diffusion mechanism came from observations of dropouts at energies of about 500 eV that were not accompanied by precipitating electrons, eliminating EMIC wave scattering as a driver of the observed loss[12]. However, it remained unclear whether higher energy electrons may still be predominantly scattered by EMIC waves.

A recent study has shown that EMIC waves are required to reproduce an unusually narrow remnant belt at energies >4 MeV[13] that was observed on the Van Allen Probes and referred to as the 'storage ring'[14]. Observations of electron pitch angle distributions also showed telltale signatures of interactions of ultra-relativistic electrons with EMIC waves[15] but did not provide observational evidence that EMIC waves can initialize the net loss at all pitch angles and not just change the shape of the pitch angle distribution.

Identifying and separating the effects of different acceleration and loss mechanisms is often a challenging task. Multiple competing acceleration and loss mechanisms usually occur simultaneously. Another complication is that acceleration and loss mechanisms also depend on the seed population of electrons at somewhat lower energy than relativistic or ultra-relativistic energies. At times when seed population fluxes are increased, all acceleration and loss mechanisms will become intensified and distinguishing between them is challenging.

Careful selection of particular conditions during specific geomagnetic storms can help isolate different acceleration and loss mechanisms, allowing us to determine the dominant processes and ultimately provide insight into the physics driving radiation belt acceleration and loss.

Up until recently, there were no accurate and reliable measurements of ultra-relativistic electron populations with full pitch angle resolution. The relativistic electron–proton telescope (REPT) instrument on the Van Allen Probes spacecraft mission[16] has provided very detailed information on the pitch angle distributions and the orbit of the spacecraft allows for the measurement of electrons trapped near the geomagnetic equator. However, for a number of storms, the flux of >3 MeV electrons was below the noise level of the high-energy channels of the REPT instrument, and energy and pitch angle distributions could not be inferred for these energies.

Evidence for EMIC wave scattering can come from observations of the electron energy spectrum, which only produces loss of electrons above a certain minimum energy as only very energetic electrons can be in resonance with these waves. Another typical signature of EMIC scattering is the narrowing of the normalized electron pitch angle distribution toward 90° (perpendicular to the local magnetic field), as only electrons with a large parallel energy and consequently small pitch angles can be in resonance with the waves.

Here we report observations of electron flux at different energies and observations of electron pitch angle distributions during 17 January 2013 storm, which are consistent with EMIC wave-induced loss for ultra-relativistic electrons, whereas electrons at relativistic energies are not affected by EMIC wave scattering. Comparison of modelling results to the observations provides further support of our conclusions. The presented observations and modelling show that the ultra-relativistic electron population experiences different loss mechanisms than the relativistic population.

## Results

**Observations during the 17 January 2013 storm.** The environment close to the Earth (radial distances <3–6 Earth radii ($R_E$)) is usually occupied by the cold plasma bubble co-rotating with the Earth, which is referred to as the plasmasphere. Inside the plasmasphere, relativistic electrons usually decay on the scale of 1–10 days due to scattering by whistler-mode plasma waves known as plasmaspheric hiss. Ultra-relativistic electrons can persist for a very long time and scattering by plasmaspheric hiss will be much slower at these energies[13].

The October 2012 geomagnetic storm[2,3], which occurred 101 days before our considered event, produced an abundance of electrons at energies of Mega-electron Volt (MeV) and multi-MeV. Elevated fluxes of these most energetic electrons in the heart of the outer belt at levels above REPT instruments' noise floor allowed for fully resolved measurements of pitch angle distributions as a function of energy. After the October storm, the ultra-relativistic (above ~4 MeV) remnant belt slowly diffused inward to L between ~3 and 4 (L is the distance from the centre of the Earth to a given magnetic field line in the equatorial plane measured in $R_E$), where the ultra-relativistic electrons cannot be significantly affected by the loss to the magnetopause and outward radial diffusion, and scattering by hiss inside the plasmasphere is weak.

During a moderate storm on 17 January (Fig. 1a,b shows the index of geomagnetic activity Kp inferred from the fluctuations of magnetic field on the ground), flux of relativistic electrons at

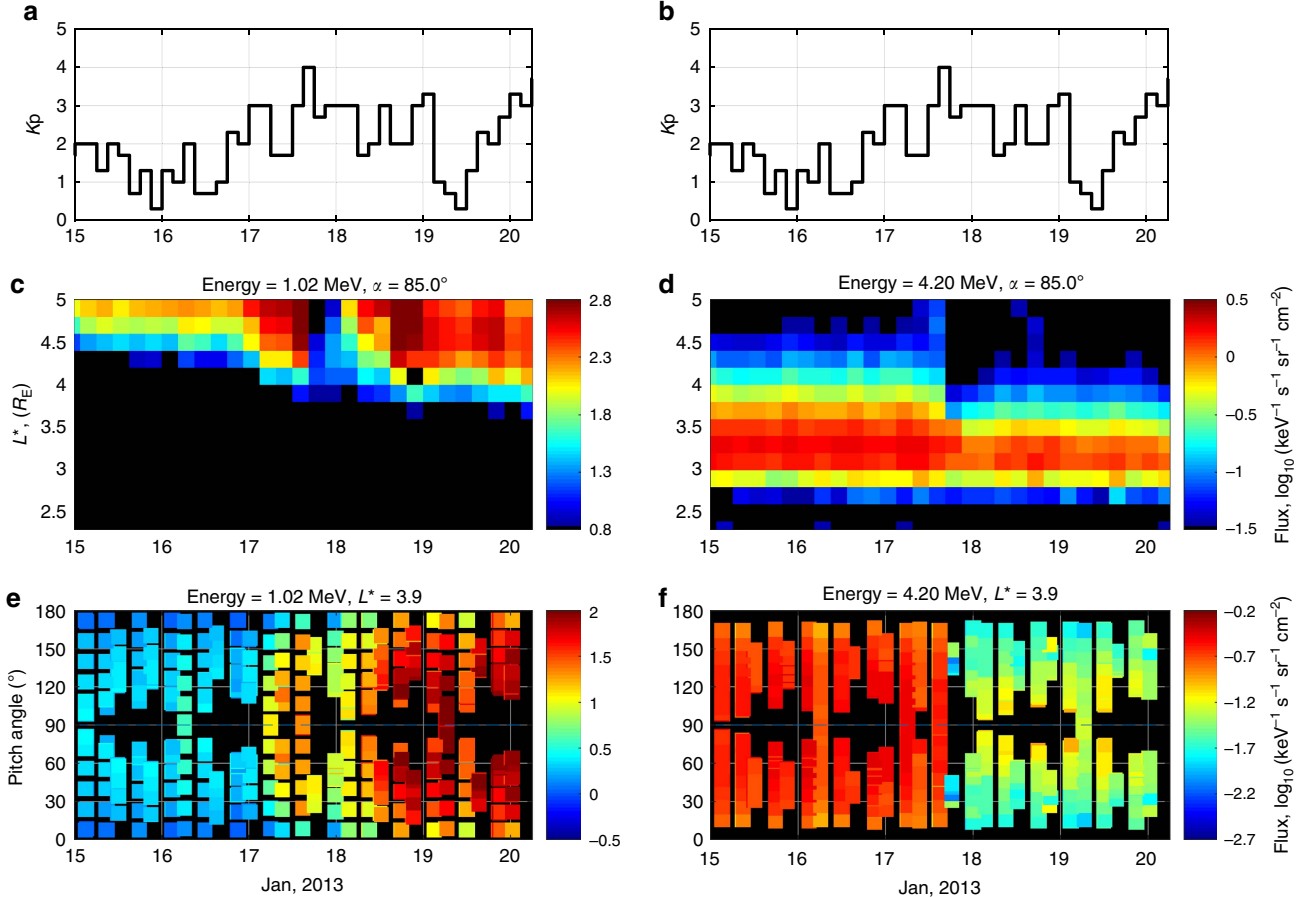

**Figure 1 | Observations of radial profiles of electron fluxes and pitch angle distributions during the 17 January 2013 storm.** (**a,b**) Evolution of the index of geomagnetic activity Kp as a function of day of January 2013. Observations of electron flux at 85° equatorial pitch angle as a function of radial distance and day at (**c**) 1.02 and (**d**) 4.20 MeV energy by the MagEIS and REPT instruments on the Van Allen Probes spacecraft. Observations of electron flux at $L^* = 3.9$ as a function of equatorial pitch angle and day by MagEIS and REPT for (**e**) 1.02 and (**f**) 4.20 MeV electrons. The radial profiles of fluxes show that over the course of the storm at 1.02 MeV electron fluxes increase in the heart of the belt (**c**), while at 4.2 MeV electrons drop out near $L = 4$ (**d**). Pitch angle distributions show narrowing at 4.2 MeV (**f**) that is not observed at 1.02 MeV (**e**).

1.02 MeV increased (Fig. 1c). The combination of local acceleration and inward radial diffusion moved the inner boundary of the outer belt to lower radial distances (Fig. 1c). The short-lived dropout of fluxes is associated with the reversible changes described in the introduction. As clearly seen in the observations after the adiabatic dropout (see Supplementary Fig. 1 and Supplementary Note 1), fluxes return to pre-storm values.

Ultra-relativistic electrons at 4.2 MeV (Fig. 1d) show a very different evolution, which is a net decrease in flux. This loss of ultra-relativistic electrons is not produced by the loss to the magnetopause, as the multi-MeV electron belt is located deep inside the outer zone, below 4.5 $R_E$, whereas the magnetopause for this event was compressed down to 7.1 $R_E$ according to an empirical model[17]( see also Supplementary Note 2 and Supplementary Figs 2 and 3) and the variation of the global magnetic field in the inner magnetosphere was not significantly large (see Supplementary Note 3 and Supplementary Fig. 4). Such difference in the evolution of fluxes between 1.02 and 4.2 MeV can be explained by the presence of scattering by EMIC waves that affects only electrons above certain minimum threshold energy, which is the minimum energy for which electrons can resonantly interact with EMIC waves.

Another clear piece of evidence for the EMIC wave scattering is provided by observation of the electron pitch angle distributions (Fig. 1e,f). During resonant interactions with EMIC waves, only small pitch angle electrons are scattered into the atmosphere and the distribution should have characteristic signatures with bite-outs near 0° and 180°, corresponding to particles with very high velocity parallel to the magnetic field that were scattered by EMIC waves. Although the pitch angle distribution of the 1.02 MeV electrons is rather wide (Fig. 1e), the ultra-relativistic electron distribution dramatically changes after the flux drop out and becomes very narrow, with sharp gradients around 50° and 130°.

EMIC waves are extremely strong plasma wave emissions that are often observed only in a very narrow localized region of magnetic local time (MLT). Although satellite measurements can only provide information at a particular radial distance and MLT, ground-based arrays of observations of EMIC waves provide a global view of waves.

Observations from an array of stations in Finland show strong EMIC emissions on 17 January, between ~15 and 18 universal time (see Fig. 2), with a peak intensity in the heart of the outer zone, exactly at the same time when the Van Allen Probes observed a dropout in ultra-relativistic fluxes (Fig. 1d) and prompt narrowing of the pitch angle distributions (Fig. 1f). The clear signature of narrowing of the distributions during the days when EMIC wave activity was seen on the ground clearly show that the pitch angle distributions are produced by resonant scattering with EMIC waves.

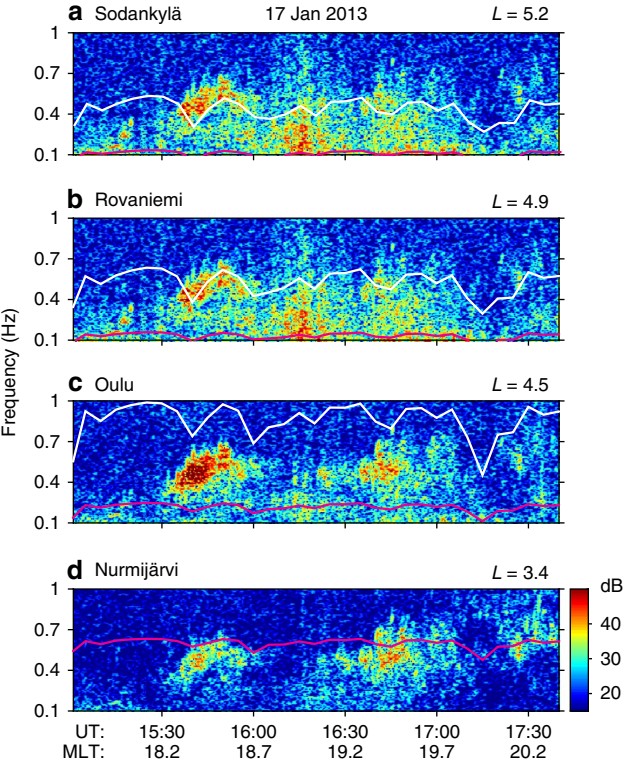

**Figure 2 | Observations of EMIC wave activity on 17 January 2013.**
Observations of EMIC wave activity on 17 January 2013 from a latitudinal
array of ground-based search-coil magnetometer stations in Finland,
arranged from highest to lowest *L*. (**a**–**d**)The spectrograms show the
observed wave power as a function of frequency and universal time (UT) on
17 January 2013 with the MLT of the stations also indicated. EMIC waves
were observed for several hours and the peak wave intensity was observed
at Oulu station at $L = 4.5$. The white line is the estimated helium
gyrofrequency and the magenta line is the estimated oxygen gyrofrequency
at the equatorial crossing of the field line passing through the station. These
lines bound the wave observations at Oulu, indicating that the source region
for the waves was likely located in the heart of the radiation belts[27]
(see Supplementary Note 4).

**Comparison of modelling with observations**. To model the
evolution of fluxes, we use numerical simulations that include
diffusion in radial distance, energy and pitch angle. The
numerical code is described in detail in the Methods section.
Simulations of relativistic electron radiation belt dynamics
including radial transport, and local acceleration and loss due to
whistler-mode waves (Fig. 3a) can well reproduce the dynamics
of the 1.02 MeV electrons (Fig. 3e). Simulation of 4.2 MeV electrons
with the same numerical code also predicts increase in fluxes
(Fig. 3b), contrary to observations that clearly show the dropout
as discussed above and shown for comparison in Fig. 3f. The
introduction of EMIC waves in the simulation during 15–18
universal time on 17 January does not significantly change the
dynamics of the 1.02 MeV electrons (Fig. 3c) but dramatically
changes the evolution of ultra-relativistic electrons. Inclusion of
EMIC waves at the time when they are observed on the ground
produces a dropout in fluxes at 4.2 MeV (Fig. 3d), consistent
with the evolution of radial profiles of fluxes observed on the
Van Allen Probes (Fig. 3f). Results of the simulations for energy
channels of 0.46 and 3.4 MeV electrons also agree well with
observations and are presented in Supplementary Fig. 5.

At 1.02 MeV, evolution of the electron pitch angle distribution
not including (Fig. 4a) and including (Fig. 4c) the effects of EMIC

waves are similar, and both simulations produce a wide pitch
angle distribution similar to the pitch angle distribution observed
on Van Allen Probes, with the exception of the adiabatic changes
(see also Supplementary Notes 1 and 2, and Supplementary Figs 2
and 3). The modelled distributions are rather wide. For the
4.2 MeV electrons, similar wide pitch angle distributions are seen
when EMIC waves are not taken into account (Fig. 4b). However,
the inclusion of EMIC waves (Fig. 4d) produces bite-outs at small
pitch angles (near 0° and 180°) and particles are transported
from high pitch angles (near 90°) towards the loss cone by
whistler-mode waves. The scale of the dropout at $L = 3.9\,R_E$ and
the narrow shape of the pitch angle distribution observed on
Van Allen Probes (Fig. 4f) is well reproduced by the simulations
with EMIC waves (Fig. 4d). The modelled pitch angle
distributions at energies of 0.46 and 3.4 MeV also agree with
observations and are presented in Supplementary Fig. 6.

## Discussion
The 17 January 2013 geomagnetic storm provided unique
conditions that allowed us to show that EMIC-induced wave
scattering can produce a very fast loss into the atmosphere at ultra-
relativistic energy. Clear differences were observed in the dynamics
at different energies, that is, the 1.02 MeV electrons are accelerated,
while EMIC wave-induced precipitation depletes the 4.2 MeV
electron population. The narrowing of the pitch angle distribution
at 4.2 MeV is consistent with EMIC wave scattering, providing
additional confirmation of the EMIC wave-induced loss.

Simulations including EMIC waves can reproduce both an
increase in flux near 1 MeV energy and a reduction at 4.2 MeV.
During this event, an abundance of ultra-relativistic electrons
near $L = 3.5$ allowed us to observe the progression of the shape of
the pitch angle distribution. Pitch angle distributions provide
additional evidence that the loss processes in the radiation belts
are different for relativistic and ultra-relativistic energies. Ultra-
relativistic electrons at moderate pitch angles have sufficient
parallel velocity to change the sense of rotation of the left-hand
polarized EMIC waves. In the rest reference frame EMIC waves
propagating along the field line have left-hand polarization,
whereas in the reference frame of the ultra-relativistic particles
the wave becomes right-hand polarized, which is the same as the
rotation sense of the electrons. Electrons gyrating around the field
line in the same rotation sense and with the same frequency as
waves in the electron reference frame can be in resonance with
waves and can be effectively scattered.

The presence of EMIC waves provides very efficient loss
mechanism for the most energetic electrons in the energy
spectrum and may impose an upper limit on the energy of the
trapped electron population. Similar processes may occur on the
outer planets and exoplanets. The absence of EMIC waves and/or
a relatively low, cold plasma density (due to lack of a dense
planetary atmosphere) would imply that electrons on exoplanets
may be potentially accelerated to extremely high energies, much
higher than that of in the Earth's magnetosphere, without being
scattered into the loss cone.

## Methods
**Model description.** The Versatile Electron Radiation Belt[18,19] code was used to
perform the simulations described in this manuscript. The Fokker–Plank equation
can be written on the single grid using adiabatic invariant coordinates[20]:

$$\frac{\partial f}{\partial t} = \frac{1}{G} \frac{\partial}{\partial L^*}\bigg|_{V,K} G\langle D_{L^*L^*}\rangle \frac{\partial f}{\partial L^*}\bigg|_{V,K} +$$
$$\frac{1}{G} \frac{\partial}{\partial V}\bigg|_{L^*,K} G\left(\langle D_{VV}\rangle \frac{\partial f}{\partial V}\bigg|_{L^*,K} + \langle D_{VK}\rangle \frac{\partial f}{\partial K}\bigg|_{L^*,V}\right) + \tag{1}$$
$$\frac{1}{G} \frac{\partial}{\partial K}\bigg|_{L^*,V} G\left(\langle D_{KK}\rangle \frac{\partial f}{\partial K}\bigg|_{L^*,V} + \langle D_{VK}\rangle \frac{\partial f}{\partial V}\bigg|_{L^*,K}\right) - \frac{f}{\tau}$$

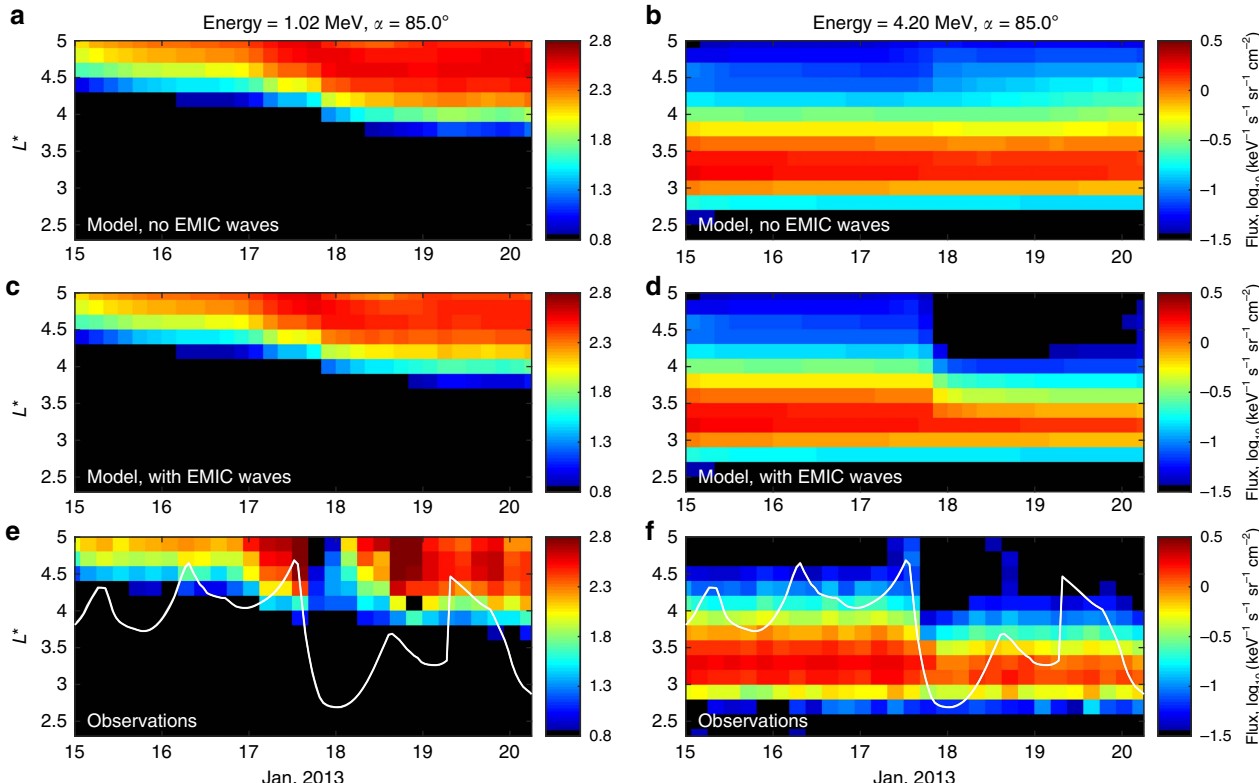

**Figure 3 | Comparisons of the evolution of modelled and observed radial profiles of electron flux.** All panels show the flux of electrons at 85° equatorial pitch angle as a function of radial distance and day of January 2013. The output of numerical simulations accounting for radial transport and scattering by VLF waves but without EMIC wave scattering are shown for (**a**) 1.02 MeV and (**b**) 4.2 MeV electrons. The output of numerical simulations similar to the first row but now with additional scattering by EMIC waves are shown for (**c**) 1.02 MeV and (**d**) 4.2 MeV electrons. Observations of electron flux from the MagEIS and REPT instruments on the Van Allen Probes spacecraft are shown for (**e**) 1.02 MeV and (**f**) 4.2 MeV electrons. The white lines in **e** and **f** show the plasmapause location. Observations show an increase in electron flux at MeV energies (**e**) and a decrease at ultra-relativistic energies (**f**). The numerical simulations including EMIC waves successfully reproduce the evolution of both of these electron populations (**c,d**).

where $f$ is the electron PSD, $L^{\star}$ is a form of the third invariant; $K = \frac{J}{\sqrt{8\mu m_0}}$, where $J$ is the second adiabatic invariant, $m_0$ is electron rest mass, $V \equiv \mu(K+0.5)^2$, where $\mu$ is the first adiabatic invariant; $\langle D_{L^{\star}L^{\star}} \rangle$, $\langle D_{VV} \rangle$, $\langle D_{KK} \rangle$ and $\langle D_{VK} \rangle$ are bounce-averaged diffusion coefficients; $G = -\frac{2\pi B_0 R_E^2}{L^{\star 2}} \frac{\sqrt{8m_0 V}}{(K+0.5)^3}$; is the Jacobean of the transformation from an adiabatic invariant system ($\mu$, $J$, $\Phi$) to ($L^{\star}$, $V$, $K$); $\Phi$ is the third adiabatic invariant, $B_0$ is the magnetic field on the surface of the Earth, $R_E$ is the Earth's radius; and $\tau$ is the electron's lifetime inside the atmospheric loss cone, which is taken to be a quarter of the bounce period. To compute radial transport, we used activity-dependent[21] radial diffusion rates. The flux of particles observed on satellites can be inferred from PSD using equation $J = f \cdot p^2$, where $j$ is the differential directional flux and $p$ is the relativistic momentum.

**Wave parameters.** Wave parameters of hiss and chorus waves are based on statistical studies and were used in the previously published long-term simulations[22,23]. Wave parameters were not specifically adjusted for this event. The simulation includes whistler-mode chorus waves on the day and night side following the Kp-based parametrizations presented in ref. 23. The chorus waves were included above the plasmapause location. Plasmaspheric hiss waves were included inside the plasmasphere based on a recent empirical model[24].

The plasmapause location was constructed based on the plasmapause test particle (PTP) simulation model[24]. In the Versatile Electron Radiation Belt code simulation, the location of the inner boundary of EMIC waves in the plasmaspheric plume and hiss in plume was calculated as the hourly average minimum of the plasmapause location calculated with the PTP model.

We assumed the presence of the plasmaspheric plume base on the PTP model[25]. The model shows clear presence of the plume during the interval from 17 January 2013, 15:00 h, to 19 January 2013, 08:00 h. The orbits of the Van Allen Probes did not cross the plume at the times when waves are observed at the ground stations in Fig. 2. However, the Van Allen Probes provide measurements of particles that drift around the Earth and are affected by waves in all MLT sectors. As discussed above, particle measurements provide clear evidence of loss by EMIC waves at ultra-relativistic energies.

The spatial distribution of the EMIC waves was assumed to be very localized at only 5% of the electron orbit. It was assumed that EMIC waves can effectively

scatter electrons only in the high-density region of the plasmaspheric plume and were included in the model only above the PTP-calculated minimum plasmapause location.

The spectral parameters of the EMIC waves were obtained from the GOES 15 spacecraft measurements. Figure 5 shows the observation of EMIC waves on 17 January 2013 and the observed spectrum. We used the superposition of three Gaussian shapes to fit the observed spectrum for the further calculations of the EMIC wave diffusion coefficients (see Supplementary Table 1). The parameters that are assumed for the calculation of the EMIC wave-induced diffusion coefficients are based on GOES measurements and are presented and discussed in the Supplementary Note 5.

For this period, plasmaspheric hiss waves in the region of the plasmaspheric plume were also included above the calculated location of the plasmapause, assuming that they affect only 5% of drift time. Corresponding diffusion coefficients were scaled by Kp index.

**Numerical approach.** The Fokker–Plank equation was solved using a fully implicit scheme on a $18 \times 101 \times 100$ point grid for the coordinates $L^{\star}$, $V$ and $K$, respectively, with a time resolution of 1 h. The 7 days of simulation from 14 January 2013 until 21 January 2013 was initialed with PSD profiles that were inferred from observation. Detailed information on how the data were processed is given in Supplementary Note 6.

The grid is uniform in $L^{\star}$ and $K$, but logarithmic in $V$ to resolve both lower energy and ultra-relativistic electrons. The boundaries for $L^{\star}$ were set up at 2.1 and 5.5, with a step of 0.2 $R_E$. The $V$ and $K$ coordinates were calculated in the range of energy and pitch angle defined at the upper radial boundary. The $K$ coordinate was calculated on a linear grid of pitch angles from 0.7° to 89.3°. At the outer radial boundary of the code, the $V$ coordinate was calculated on the logarithmic grid using the range of energies from 10 keV to 10 MeV.

The upper boundary conditions for $L^{\star}$ were reconstructed from the Van Allen Probes observations using MagEIS and REPT measurements. The lower radial boundary, the upper boundary condition for $V$ (high-energy population) and the upper boundary condition for $K$ (electrons inside the loss cone) were set to zero, to simulate the absence of the electrons on the edge of the grid (see refs 21,23 for more details). Practically, identical results were obtained if a zero gradient

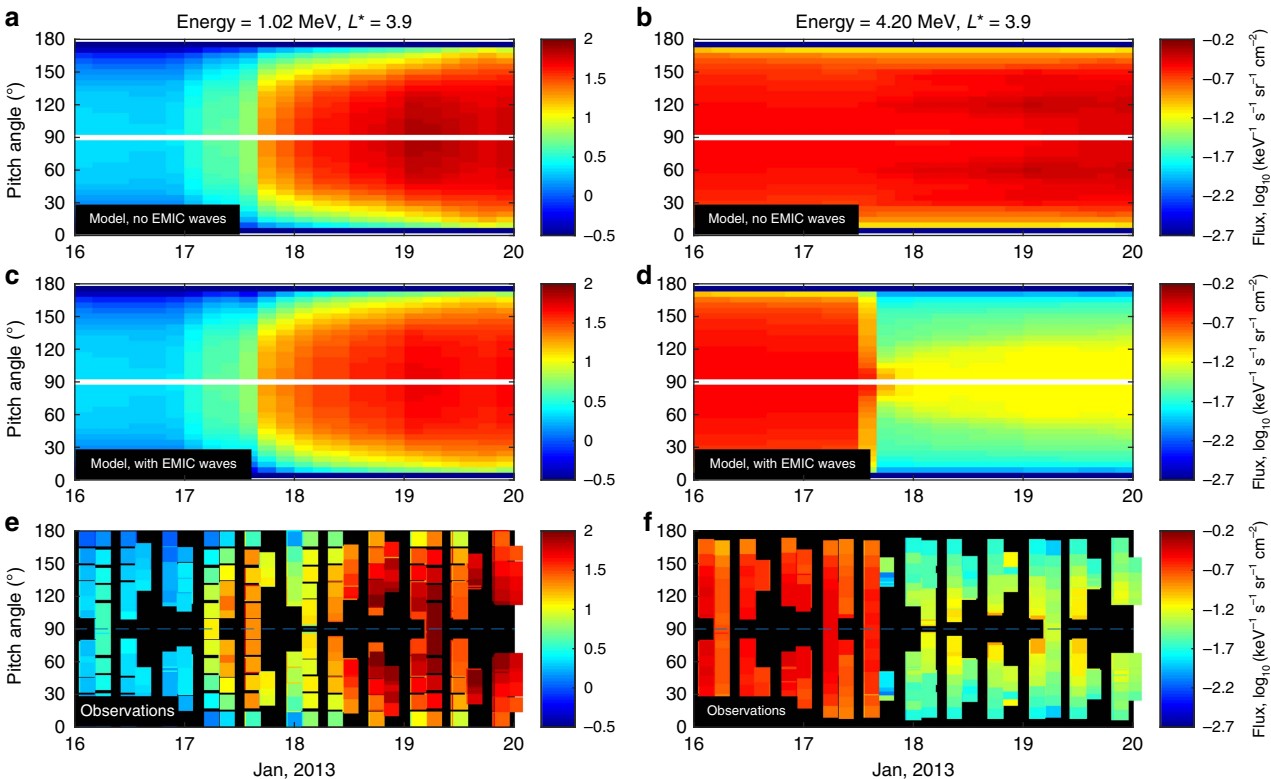

**Figure 4 | Evolution of the electron pitch angle distribution.** All panels show the flux of electrons as a function of equatorial pitch angle and day of January 2013 at $L^* = 3.9$. Results of numerical simulations accounting for radial transport and scattering by VLF waves but without EMIC wave scattering are shown for (**a**) 1.02 MeV and (**b**) 4.2 MeV electrons. Simulations including EMIC wave scattering are shown for (**c**) 1.02 MeV and (**d**) 4.2 MeV electrons. Observations from the MagEIS and REPT instruments on the Van Allen Probes spacecraft are shown for (**e**) 1.02 MeV and (**f**) 4.2 MeV electrons. Observations of the narrowing of the pitch-angle distributions during the storm (**f**) presents clear telltale signatures of EMIC wave scattering and is very similar to the modelled evolution of pitch angle distributions including EMIC wave effects (**d**).

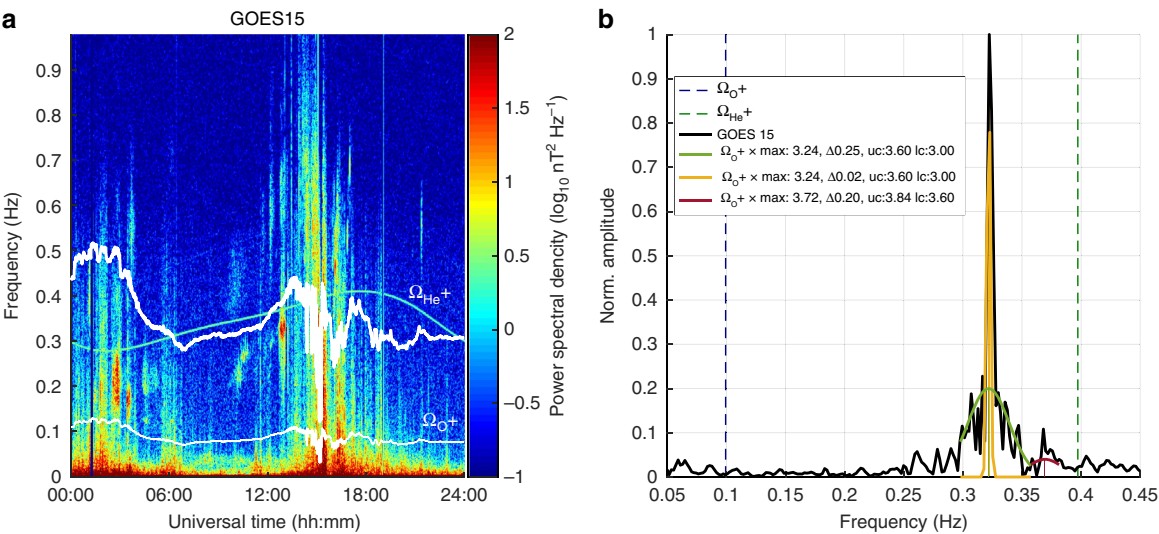

**Figure 5 | Observation of the EMIC waves on GOES 15.** (**a**) The dynamic spectrogram from GOES 15 on 17 January 2013 showing wave power spectral density as a function of frequency and universal time (UT). The bold white line shows the helium gyrofrequency and narrow white line shows the oxygen gyrofrequency. (**b**) The observed frequency spectrum of waves on 17 January 2013 at 12:55 UT (black) and three Gaussian fits to the observations (green, yellow and red). The EMIC Gaussian wave spectrum parameters were defined by a central frequency (max), frequency bandwidth ($\Delta$), lower (lc) and upper (uc) cutoff frequencies. The parameters are presented as multiples of the oxygen gyrofrequency $\Omega_{O^+}$ ( blue dashed line) and defined in Supplementary Table 1. The helium gyrofrequency $\Omega_{He^+}$ is shown by the vertical dashed green line. The wave amplitudes are also presented in Supplementary Table 1.

boundary condition is assumed at the higher $K$ boundary representing a zero gradient in the pitch angle distribution at 0°. Constant PSD was prescribed at the lower boundary condition for $V$ representing a balance between convective sources and losses. Although these conditions may not adequately represent the dynamics of the lower-energy electrons, the exact values of fluxes at this boundary does not substantially influence the results. Detailed sensitivity simulations to the assumed lower-energy boundary have been previously presented[26].

**Data availability.** All the Van Allen Probes data are publicly available at http://www.rbsp-ect.lanl.gov/data_pub/ by the REPT and MagEIS instrument. The GOES data are publicly available at http://satdat.ngdc.noaa.gov/sem/goes/data/. The OMNI data are obtained from http://omniweb.gsfc.nasa.gov/form/dx1.html. The modelling results of the plasmasphere are available on RBSP-ECT Data Portal (http://www.rbsp-ect.lanl.gov/). Kp index of geomagnetic activity was obtained from the GSFC/SPDF OMNIWeb interface at http://omniweb.gsfc.nasa.gov and produced by GFZ, Potsdam.The results of the model simulations and any other relevant data or information that may be necessary to reproduce the presented results is available from the authors upon request.

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

## Acknowledgements

This research was supported by the Helmholtz Association Recruiting Imitative programme, NASA grants NNX16AF91G,NNX13AE34G, NNX10AK99G, NNX15AI94G and NNX16AG78G, RBSP-ECT funding through JHU/APL contract 967399 under NASA contract NAS5-01072, JHU/APL contract 922613 (RBSP-EFW), NASA Grant NNX16AF85G, NSF GEM AGS-1203747 and UC Lab Fee grant 12-LR-235337, project PROGRESS funded by EU Horizon 2020 No 637302 and International Space Science Institute (Bern).

## Author contributions

Y.Y.S. initiated, conserved, led the study and coordinated the efforts among different institutions. The manuscript was written by Y.Y.S. with contributions from M.S. and other co-authors. A.Y.D. performed simulations under supervision of Y.Y.S., H.E.S., D.N.B., T.J.R. and M.J.E. provided data. M.S., M.E.U. and O.V.A. worked with wave data analysis. A.C.K., A.Y.D., I.S.Z., H.Z, N.A and O.V.A. worked with particle data.

## Additional information

**Competing financial interests:** The authors declare no competing financial interests.

