## [Peer Review File · Nature Communications]

Reviewer #1 (Remarks to the Author):

This is a very clear study of a good example where the effects of EMIC waves on very high energy electrons can be isolated. The analysis is thorough, and the methods are clear.

This study will make a contribution to radiation-belt physics and to future calculations of space weather.

I believe it is worthy and ready for publication in Nature Communications.

Two minor typos:

1. In the Methods section, the 4th paragraph begins "We assumed presence", it should read "We assumed the presence".

2. The second from last paragraph of the Methods section begins " The grid is uniform in L^* and K , but logarithmic in K ". Should this be "The grid is uniform in L^* and K , but logarithmic in V " ?

Reviewer #2 (Remarks to the Author):

Dear Authors

The paper, "Wave-Induced Loss of Ultra-Relativistic Electrons in the Van Allen Radiation Belts" touches hot topics in the radiation belt physics. It was really my pleasure to review this paper. I think this paper was written well. However, unfortunately I cannot easily agree to the main ideas of this paper as following reasons.

1. This paper insists that ultra-relativistic electron flux decrease by EMIC wave scattering, while relativistic electrons experience reversible adiabatic dropout in January 17 event.

In my eye, the flux dropout of all energy electrons occurred simultaneously. If only ultra-relativistic electron dropout was observed or there was time difference between relativistic and ultra-

relativistic electron dropouts, I would be a strong supporter. If something happen at the same time, it is natural that these events have same origin. The authors pointed out fast flux recovery of relativistic electron as an evidence of adiabatic dropout. However, I think a loss process following fast acceleration also explains this dropout event.

2. I think the authors only consider two loss processes, magnetopause shadowing and EMIC wave scattering. We cannot say if it is not magnetopause shadowing, then it should be EMIC scattering. I have studied of electron precipitation for a long time and found a lot of different type precipitation structures. This means a number of loss processes, even we don't fully understand, are related to the electron loss. Regarding the first argument, any other mechanism cannot be excluded as a candidate of electron dropout occurred on January 17. So, I cannot agree the energy spectrum behavior observed by VAP mission is the direct evidence of EMIC wave scattering.

3. In figure2, EMIC wave activity observed in L=3.3 where the 4.2MeV electron flux did not decrease significantly in VAP observation. It seems to need some explanation.

4. The authors pointed out that another evidence of EMIC scattering is pitch angle narrowing. It might be true. However, the pitch angle distribution could be changed by magnetic field variation during drift motion, where third adiabatic invariant is not conserved. NOAA GOES satellite magnetometer data shows significant magnetic field disturbance on January 17, 2013. In order to say the pitch angle distributions are resulted from the EMIC wave interaction, the authors should show global scale magnetic field variation, third adiabatic invariant violation could not generate such pitch angle distribution.

Minor comment

In figure 1, the Maximum Kp index was just 4 for only 3 hours, many readers might not agree to the word, "relatively strong storm" shown in this article.

Reviewer #3 (Remarks to the Author):

Wave-Induced Loss of Ultra-Relativistic Electrons in the Van Allen Radiation Belts

A) This study showed that rapid scattering of ultra-relativistic electrons by EMIC waves which are different from relativistic electron loss. Computer simulations which can be compared with Van

Allen Probes observation clearly showed that EMIC waves are essential for ultra-relativistic electron precipitations.

B) Although there have been several reports to show wave-induced loss of ultra-relativistic electrons (e.g., reviewed by Turner et al., Outer radiation belt flux dropouts; Current understanding and unresolved questions, AGU monograph, 2012), this is the first clear evidence from the equatorial satellite.

C) Data and methods are appropriate.

D) Uncertainties on the simulation have been discussed.

E) About the conclusion from the simulation, I have question on the simulation.

Figure 2 provides Pc1 data at several ground-based stations. Could you provide the cyclotron frequency of protons at the magnetic equator for each L-shell? The proton cyclotron frequency provides information what L-shells are source region of EMIC waves.

Reference; Sakaguchi et al., Simultaneous ground and satellite observations of an isolated proton arc at subauroral latitudes, *J. Geophys. Res.*, 112, doi:10.1029/2006JA012135, 2007.

F) I would like to suggest to specify L-shell distribution of EMIC waves for the simulation, by comparing with Figure 2.

G) References are appropriate.

H) Abstract/summary are appropriate.

Reviewers' comments:

Reviewer #1 (Remarks to the Author):

We would like to thank the reviewer for the careful reading of the manuscript. We have corrected the two minor typos identified by the reviewer.

This is a very clear study of a good example where the effects of EMIC waves on very high energy electrons can be isolated. The analysis is thorough, and the methods are clear.

This study will make a contribution to radiation-belt physics and to future calculations of space weather.

I believe it is worthy and ready for publication in Nature Communications.

Two minor typos:

1. In the Methods section, the 4th paragraph begins "We assumed presence", it should read "We assumed the presence".
2. The second from last paragraph of the Methods section begins " The grid is uniform in L^* and K , but logarithmic in K ". Should this be "The grid is uniform in L^* and K , but logarithmic in V " ?

Reviewer #2 (Remarks to the Author):

Dear Authors

We would like to thank the reviewer for the careful reading of the manuscript. Thank you for your thoughtful comments. In this resubmission we are now providing additional supporting information which we think makes even better case than the 1st submission. We hope that reviewer will now find sufficient supporting information.

The paper, "Wave-Induced Loss of Ultra-Relativistic Electrons in the Van Allen Radiation Belts" touches hot topics in the radiation belt physics. It was really my pleasure to review this paper. I think this paper was written well. However, unfortunately I cannot easily agree to the main ideas of this paper as following reasons.

1. This paper insists that ultra-relativistic electron flux decrease by EMIC wave scattering, while relativistic electrons experience reversible adiabatic dropout in January 17 event.

In my eye, the flux dropout of all energy electrons occurred simultaneously. If only ultra-relativistic electron dropout was observed or there was time difference between relativistic and ultra-relativistic electron dropouts, I would be a strong supporter. If something happen at the same time, it is natural that these events have same origin. The authors pointed out fast flux recovery of relativistic electron as an evidence of adiabatic dropout. However, I think a loss process following fast acceleration also explains this dropout event.

It is common for the radiation belt that number of different processes occur during storms. As mentioned in the paper the presented event is unique and allows us to clearly separate the processes.

On January 17th fluxes drop almost instantaneously during storm and then recover approximately a day later within a few hours. Below are four complimentary evidences that this dropout is adiabatic at energies of approximately 1-2 MeV. Pitch angel distributrions show that variations at 1 MeV are not produced by EMIC waves.

- 1) Fast dropout and a prompt recovery to prestorm values are typical signatures of adiabatic changes as fluxes recover promptly to prestorm values. Observations on GOES clearly confirm that (See figure

below) Electrons and Ions at all energies drop simultaneously when magnetic field decreases and recover to prestorm values when the magnetic field increases back. This time period corresponds to the temporal dropout on Figure 1c. Note that higher multi-MeV electrons that are shown in the main part of the manuscript do not recover to the prestorm values.

Figure (also Supplementary figure 1) Goes observations of a) total magnetic field, b) differential proton fluxes from MAGPD, c) differential proton fluxes from EPEAD, d) differential electron fluxes from MAGED, and e) integral electron fluxes from EPEAD. A similar variation in electron and proton fluxes are observed over the January 17th to January 19th interval shown across keV to MeV energies. Note that 2 MeV reached the instrumental noise floor level which explains flat plateau from ~1700 on Jan 17th until 4am on Jan 18th. Ultra-relativistic energy electrons are not available on GOES.

- 2) There is a clear evidence that the variations are not produced by the loss to the magnetopause as the magnetopause is at higher radial distances (Supplementary Figure 2), and pitch angle distributions do not show butterfly shapes (Supplementary Figure 3).
- 3) It is difficult to explain that fast dropout and build-up of electrons by non-adiabatic loss followed by non-adiabatic acceleration. Non-adiabatic acceleration of relativistic electrons takes several days and observations on Van Allen Probes do not show strong chorus or hiss waves that would explain the observed fast loss or fast acceleration, see the overview of waves below.

Figure, Wave spectrograms from Van Allen Probes showing observed VLF waves during January 17th, 2013. The strongest waves are on the scale of tens of pT 10^{-2} nT.

Please note that we do that into account scattering by hiss and chorus. Van Allen Probes observations do now show any unusually strong waves that can resonate with MeV or Multi-MeV particles. The square of the amplitude of these waves, which determines the scattering rates, is 10^3 - 10^6 times larger for EMIC waves than for typical hiss or chorus waves.

- 4) Excellent comparison of modelling with observations provides additional evidence that EMIC waves are responsible for the dropout at multi - MeV energies while not effecting MeV energies.
- 5) Most importantly, pitch angle distributions at relativistic energies show bite outs at small pitch angles typical for EMIC waves while lower energy electrons do not show this characteristic bite out.

2. I think the authors only consider two loss processes, magnetopause shadowing and EMIC wave scattering. We cannot say if it is not magnetopause shadowing, then it should be EMIC scattering. I have studied of electron precipitation for a long time and found a lot of different type precipitation structures. This means a

number of loss processes, even we don't fully understand, are related to the electron loss. Regarding the first argument, any other mechanism cannot be excluded as a candidate of electron dropout occurred on January 17. So, I cannot agree the energy spectrum behavior observed by VAP mission is the direct evidence of EMIC wave scattering.

To our knowledge, there is no other mechanism that has ever been suggested that can produce the observed signatures of energy distribution (affecting only energies above several MeV) and pitch angle distribution (bite outs at small pitch angles observed only for energies of above several MeV). Comparison of modelling and observations provided additional evidence that EMIC waves are responsible for the loss of multi-MeV electrons. We include in our model scattering by hiss and chorus and neither of these loss mechanisms can explain the observed behavior of the belts.

3. In figure 2, EMIC wave activity observed in L=3.3 where the 4.2 MeV electron flux did not decrease significantly in VAP observation. It seems to need some explanation.

For specific event studies, satellite observations may not be available at locations where waves are present. Often ground-based observations of the EMIC waves are used since there exists a multitude of magnetometer arrays spread in longitude about the globe. However, due to various wave propagation effects such as ionospheric ducting, EMIC waves that originate near the equatorial plane at a single L shell may be observed on the ground over a much wider range of L. It is also useful to estimate the proton, helium and oxygen cyclotron frequencies in the equatorial magnetosphere and compare these frequencies to the observed frequencies of the waves.

Waves observed at the surrounding stations have a similar frequency time structure but lower intensity. Ground observations cannot provide us exact range of radial distances but it's likely that the peak intensity was near 4.5 RE. Due to propagation effects the EMIC waves may be observed at both the higher and lower latitude stations, which explains the signal observed at the station located at L=3.4.

This is also confirmed by examining the helium cyclotron frequency (white) and the oxygen cyclotron frequency (magenta) calculated by tracing the field line to the magnetic equator using the T04s model and prevailing solar wind and geomagnetic conditions (see modified Figure 2). The proton cyclotron frequency is above the limits of the plot for all panels. Here we see that only at Oulu do the wave fall distinctly into the helium band between the two cyclotron frequencies. Waves observed at Nurmijarvi map to below the helium band and likely were not generated at these low L values.

We added a supplementary note on interpretation of the ground based observations to the supporting material .

4. The authors pointed out that another evidence of EMIC scattering is pitch angle narrowing. It might be true. However, the pitch angle distribution could be changed by magnetic field variation during drift motion, where third adiabatic invariant is not conserved. NOAA GOES satellite magnetometer data shows significant magnetic field disturbance on January 17, 2013. In order to say the pitch angle distributions are resulted from the EMIC wave interaction, the authors should show global scale magnetic field variation, third adiabatic invariant violation could not generate such pitch angle distribution.

We believe that this is the primary concern of the reviewer, and by addressing it, we hope to convince the reviewer that EMIC waves provide the explanation for the observed behavior of fluxes. To make a significant change in the distribution and produce the observed bite outs at small pitch angles, the change in the magnetic field would need to be comparable to the background magnetic field. Note that at L~4 the magnetic field strength is almost 5 times stronger than at geosynchronous orbit. As reviewer pointed out, this storm is only moderate. The modeled fluctuations of the magnetic field by the most accurate TS07 D shows that fluctuations are very small for this storm and for these radial distances. We added Supplementary Figure 4 to show the

mean change in magnetic field, normalized by the magnetic field at several radial distances from the Earth's surface. The only significant variation in the global B-field occurred earlier in the day, and that variation actually overestimated the measured variation from the EMFISIS instruments. The global B-field variation is shown in the first figure below, while the period with the largest B-field variation is shown in the second figure, with the measured B-field over plotted as thick solid lines for RBSPa and RBSPb. One can see that the global field variation was slow for this period, especially at 15-18 UT when the EMIC waves were observed. Hence, global field variations cannot explain the observed dropout and narrowing of the electron pitch angle distribution at high energies.

Figure "MLT-averaged magnetic field variation at several R values in the TS07D field model. The variation at each R is shown by each solid colored line, color-coded as shown in the legend. The left panel shows the variation over the entire day of 20130117, while on the right the variation is shown just from 10-12 UT on the same day. Overplotted on the right are EMFISIS observations for the same interval. It is shown that, although the TS07D model indicated a large ~20% global variation in the magnetic field, no such variation was observed on the Van Allen Probes, which were near apogee $\sim R = 5$ at the same time.

Moreover the changes in the pitch angle distribution were produced by the magnetic field that would affect all energies MeV and multi MeV. Observations clearly show bite outs only at multi-MeV energies and not at MeV energies.

Reply

Minor comment

In figure 1, the Maximum Kp index was just 4 for only 3 hours, many readers might not agree to the word, "relatively strong storm" shown in this article.

Thank you we corrected to moderate storm.

Reviewer #3 (Remarks to the Author):

We would like to thank the reviewer for the careful reading of the manuscript.

Wave-Induced Loss of Ultra-Relativistic Electrons in the Van Allen Radiation Belts

A) This study showed that rapid scattering of ultra-relativistic electrons by EMIC waves which are different from relativistic electron loss. Computer simulations which can be compared with Van Allen Probes observation clearly showed that EMIC waves are essential for ultra-relativistic electron precipitations.

B) Although there have been several reports to show wave-induced loss of ultra-relativistic electrons (e.g., reviewed by Turner et al., Outer radiation belt flux dropouts; Current understanding and unresolved questions, AGU monograph, 2012), this is the first clear evidence from the equatorial satellite.

C) Data and methods are appropriate.

D) Uncertainties on the simulation have been discussed.

E) About the conclusion from the simulation, I have question on the simulation.

Figure 2 provides Pc1 data at several ground-based stations. Could you provide the cyclotron frequency of protons at the magnetic equator for each L-shell? The proton cyclotron frequency provides information what L-shells are source region of EMIC waves.

We would like to thank the reviewer for this very helpful suggestion. We now provide the cyclotron frequencies and have added the reference to Sakaguchi et al., Thank you for this suggestion. It provides additional evidence that the waves are present in the heart of the outer zone.

Reference; Sakaguchi et al., Simultaneous ground and satellite observations of an isolated proton arc at subauroral latitudes, J. Geophys. Res., 112, doi:10.1029/2006JA012135, 2007.

F) I would like to suggest to specify L-shell distribution of EMIC waves for the simulation, by comparing with Figure 2.

It may be difficult to exactly pinpoint the minimum L-shell where the waves are present from the ground observations. Reviewer's suggestion certainly helped us. The approximate location is consistent with our assumption. The Oulu station is at 4.5 is always outside of the predicted by the PTP plasmopause location where we assume that waves are present, and Nurmijarvi station is just inside of the region where we assume the waves are present. We now explicitly discuss in the methods section how we calculate the location of the EMIC waves based on the PTP model. The assumed location of waves is variable but consistent with the observations of waves.

G) References are appropriate.

H) Abstract/summary are appropriate.

Reviewer #2 (Remarks to the Author):

Dear Authors

I have reviewed the revised article and now I believe most of my comments are cleared. Here, I would like to give you some minor comments that might be remained unsolved issues.

1. With respect to my first comment, the authors show fast electron flux dropout and recovery that is coupled by geomagnetic field fluctuations. Such electron flux variation implies the dropout is produced by adiabatic motion. However, ultra-relativistic electrons looks remain at low flux density that implies the electrons lost permanently. In order to confirm this, I recommend to calculate phase space density of the electrons.

2. Regarding my second comment, while there are lots of different type electron precipitations have been observed, just a few papers have reported about them and I agree such loss mechanism has not been studied well. Mabey in future, new space missions will reveal such complex process of such particle motions.

3. The third comment is explained by difference between ground and space observations. I agree with authors, I mean we may not exactly understand how EMIC waves propagate on ground. I recommend to include short description that ground observations could be different from the space observations in the manuscript.

4. On the fourth comment, the author's opinion that small magnetic field fluctuation does not make signification changes in pitch angle distribution might be right. I agree with authors. To confirm this, we need to statistical study of pitch angle changes. I know there are some reports studying low energy electron's pitch angle distributions but ultra-relativistic electrons. This could be future work.

Reviewer #3 (Remarks to the Author):

Wave-Induced Loss of Ultra-Relativistic Electrons in the Van Allen Radiation Belts

The authors revised their manuscript along my comments. I can now recommend this paper for publication in Nature Communications.

Reply to reviewer's comments is marked as blue bold.

I have reviewed the revised article and now I believe most of my comments are cleared.

We are glad that the reviewer found our respond convincing.

Here, I would like to give you some minor comments that might be remained unsolved issues.

1. With respect to my first comment, the authors show fast electron flux dropout and recovery that is coupled by geomagnetic field fluctuations. Such electron flux variation implies the dropout is produced by adiabatic motion. However, ultra-relativistic electrons looks remain at low flux density that implies the electrons lost permanently.

We are glad that we convinced reviewer that the variations at MeV are adiabatic while at multi-MeV the loss is irreversible.

In order to confirm this, I recommend to calculate phase space density of the electrons.

Below is the plot of phase space density that is consistent with the localized loss at multi-MeV in the heart of the belts. The minimum around L=4 is consistent with the localized loss by EMIC waves as discussed in the paper.

2. Regarding my second comment, while there are lots of different type electron precipitations have been observed, just a few papers have reported about them and I agree such loss mechanism has not been studied well. Maybe in future, new space missions will reveal such complex process of such particle motions.

As we mentioned in our previous communications the only mechanism that has been suggested so far that produce such signatures in the energy flux and pitch angle distributions is EMIC-induced wave scattering.

3. The third comment is explained by difference between ground and space observations. I agree with authors,

I mean we may not exactly understand how EMIC waves propagate on ground. I recommend to include short description that ground observations could be different from the space observations in the manuscript.

Supplementary Note 4, entitled “On interpretation of ground observations” discusses in details how EMIC waves propagate and how to interpret ground observations.

4. On the fourth comment, the author's opinion that small magnetic field fluctuation does not make significant changes in pitch angle distribution might be right. I agree with authors. To confirm this, we need to statistical study of pitch angle changes. I know there are some reports studying low energy electron's pitch angle distributions but ultra-relativistic electrons. This could be future work.

We are glad that reviewer agrees with us that magnetic field fluctuations do not make significant changes in the pitch angle distributions.